# Field assessment of BinaxNOW antigen tests as COVID-19 treatment entry point at a community testing site in San Francisco during evolving omicron surges

John Schrom[1], Carina Marquez[1], Chung-Yu Wang[2], Aditi Saxena[2], Anthea M. Mitchell[2],
Salu Ribeiro[3], Genay Pilarowski[2], Robert Nakamura[4], Susana Rojas[3], Douglas Black[1],
Maria G. Contreras Oseguera[3], Edgar Castellanos Diaz[3], Joselin Payan[3], Susy Rojas[3],
Diane Jones[3], Valerie Tulier-Laiwa[5], Aleks Zavaleta[5], Jacqueline Martinez[3],
Gabriel Chamie[1], Carol Glaser[4], Kathy Jacobson[4], Maya Petersen[6], Joseph DeRisi[2◉],
Diane V. Havlir[1◉*]

1 Division of HIV, Infectious Diseases, and Global Medicine, University of California, San Francisco, San Francisco, California, United States of America, 2 Chan Zuckerberg Biohub, San Francisco, California, United States of America, 3 Unidos en Salud, San Francisco, California, United States of America, 4 California Department of Public Health, Richmond, California, United States of America, 5 Unidos en Salud and Latino Task Force, San Francisco, California, United States of America, 6 Division of Biostatistics, University of California, Berkeley, Berkeley, California, United States of America

◉ These authors contributed equally to this work.
* Diane.Havlir@ucsf.edu

**Data Availability Statement:** All relevant data are within the manuscript and its Supporting Information files.

## Abstract

COVID-19 oral treatments require initiation within 5 days of symptom onset. Although antigen tests are less sensitive than RT-PCR, rapid results could facilitate entry to treatment. We collected anterior nasal swabs for BinaxNOW and RT-PCR testing and clinical data at a walk-up, community site in San Francisco, California between January and June 2022. SARS-CoV-2 genomic sequences were generated from positive samples and classified according to subtype and variant. Monte Carlo simulations were conducted to estimate the expected proportion of SARS-CoV-2 infected persons who would have been diagnosed within 5 days of symptom onset using RT-PCR versus BinaxNOW testing. Among 25,309 persons tested with BinaxNOW, 2,799 had concomitant RT-PCR. 1137/2799 (40.6%) were SARS-CoV-2 RT-PCR positive. We identified waves of predominant omicron BA.1, BA.2, BA.2.12, BA.4, and BA.5 among 720 sequenced samples. Among 1,137 RT-PCR positive samples, 788/1137 (69%) were detected by BinaxNOW; 94% (669/711) of those with Ct value <30 were detected by BinaxNOW. BinaxNOW detection was consistent over lineages. In analyses to evaluate entry to treatment, BinaxNOW detected 81.7% (361/442, 95% CI: 77–85%) of persons with COVID-19 within 5 days of symptom onset. In comparison, RT-PCR (24-hour turnaround) detected 84.2% (372/442, 95% CI: 80–87%) and RT-PCR (48-hour turnaround) detected 67.0% (296/442, 95% CI: 62–71%) of persons with COVID-19 within 5 days of symptom onset. BinaxNOW detected high viral load from anterior nasal swabs consistently across omicron sublineages emerging between January and June of 2022. Simulations support BinaxNOW as an entry point for COVID-19 treatment in a community field setting.

**Funding:** The funding for this study was provided by University of California, San Francisco, the Chan-Zuckerberg Biohub, the San Francisco Department of Public Health, the California Department of Health, and the McGovern Foundation. The BinaxNOW cards were provided by the California Department of Health. The funders had no role in study design, data collection and analysis, decision to publish, or preparation of the manuscript.

**Competing interests:** The authors have declared that no competing interests exist.

## Introduction

SARS-CoV-2 rapid antigen tests are public health tools that can be used for COVID-19 diagnosis, to identify persons most at risk for transmission, and as mileposts for isolation [1]. They effectively detect high levels of virus but are less sensitive than reverse transcriptase polymerase chain reaction (RT-PCR) assays at lower viral levels. Rapid antigen tests are used as a surrogate for infectiousness based on their correlation to in vitro cultures, although this application is imperfect with gaps in existing data [2–6]. Rapid antigen tests have the advantage of providing results quickly and outside of medical clinics or in the home [1, 7]. One limitation of these assays is that during the upswing of virus after initial infection, they detect fewer cases compared to RT-PCR testing [8]. For this reason, repeat rapid antigen testing is recommended for persons with suspected infection but a negative rapid antigen test.

There are now 2 oral treatments—nirmatrelvir/ritonavir (Paxlovid™) and molnupiravir (Lagevrio™)—for persons with COVID-19 deemed at high risk for progression to serious disease [9]. Both treatments require initiation within 5 days of COVID-19 symptom onset. Given that delay times for the return of PCR test results are often over 24 hours, rapid antigen tests could play an important role in identifying persons who could benefit from treatment; however, it remains unclear to what extent this advantage outweighs the limitations of sensitivity in early infection.

We have been operating a walk-up COVID-19 rapid test-and-respond site in the Mission District of San Francisco through an academic, community, and public health partnership that provides low-barrier services that extend the reach of traditional health systems and conducts ongoing molecular surveillance for SARS-CoV-2 [10–12]. The program includes community-led provision of education and supplies for low-income households with new COVID-19 diagnoses [13]. We have previously described the performance of the BinaxNOW rapid antigen tests in this setting, including demonstrating that rapid tests can be used as a public health tool to increase effective isolation time due to the rapid result turnaround linked to a community-led response team [11, 14].

Since January 2022, the community served at our testing site has experienced omicron surges with omicron waves of BA.1, BA.2, BA.2.12, BA.4 and BA.5 sublineages. Our primary objective was to evaluate BinaxNOW compared to RT-PCR in our community setting for case detection for treatment evaluation that requires diagnosis within 5 days of symptom onset. With reports of varying symptom presentations with COVID-19 variants and evolving population immunity, we also evaluated BinaxNOW compared to RT-PCR for case detection by time from symptom onset for SARS-CoV-2 sublineages and over the six-month period of January to June 2022.

## Materials and methods

### Study setting and testing procedures

The Unidos en Salud testing site is a walk-up, free, rapid COVID-19 testing and vaccine site located in a parking lot on the major commercial walkway in the Mission District of San Francisco, California. The site offers weekend and weekday hours and serves a predominant Latino and immigrant community with a high proportion of frontline workers and multi-generational families that have experienced a disproportionate burden of COVID in San Francisco [15–17]. The site is led by an academic (University of California, San Francisco and Chan Zuckerberg Biohub), community (Latino Task Force), and public health (San Francisco Department of Public Health) partnership.

Prior to testing, persons provide demographic characteristics, symptom data, vaccine status, and informed consent. Certified laboratory assistants collect bilateral anterior nasal swabs for BinaxNOW according to manufacturer specification. They collected a second swab for RT-PCR in a DNA/RNA Shield (Zymo research) [18]. To assess BinaxNOW performance, we submitted samples for RT-PCR on all specimens during two periods: January 3 to 14 ("January") and May 31 to June 26 ("June"). Beyond these periods, RT-PCR was submitted for sequencing for BinaxNOW positive samples only as part of ongoing surveillance. Certified readers with oversight by a quality control manager read BinaxNOW cards. We returned results to clients using secure messaging in the Primary Health platform within an hour. Persons with a positive test were contacted within 2 hours by bilingual staff to provide support services.

### SARS-CoV-2 genomic sequencing

As previously described, we performed RT-PCR using N and E gene probes on a random sample of the nasal swabs in the DNA/RNA shield with a positive human control (RNase P) [18]. The assay limit of detection is 100 viral copies per milliliter. Cycle threshold below 40 are considered positive. For samples in January, March, April, May, and June 2022, the Illumina platform (NextSeq/NovaSeq) and the ARTIC Network V3 amplicon strategy were used for full viral genome sequencing as described [19]. Consensus genomes were assembled using the freely available CZID pipeline (czid.org), and variant lineages were called using the Pangolin Lineage Assigner (https://pangolin.cog-uk.io). Complete genomes were deposited in GISAID (virus name "hCoV-19/USA/CA-UCSF-JDxxxx/2022").

### Analyses

We calculated the proportion of persons testing BinaxNOW positive among all persons testing positive on RT-PCR and stratified analyses by SARS-CoV-2 lineage and time from symptom onset. Monte Carlo simulations were conducted to estimate the proportion of participants that would be identified as positive within five days of symptom onset based on RT-PCR versus BinaxNOW testing under varying assumptions on RT-PCR turnaround time and probability of BinaxNOW repeat testing. The population for this analysis included symptomatic participants positive for SARS-CoV-2 by RT-PCR in January and June whose symptom onset was within five days of testing. RT-PCR turnaround time was simulated using the exponential distribution; probability of BinaxNOW repeat testing in 3 days for participants initially testing negative was simulated using the binomial distribution. For the main analysis, 10,000 bootstrapped samples were taken using an expected RT-PCR return time of 1 day and expected BinaxNOW repeat testing probability of 15%. Sensitivity analyses were conducted by varying the expected RT-PCR return time (0.04, 0.5, 1, 1.5, 2 days) and varying the expected BinaxNOW return probability (0%, 25%, 50%, 75%, 100%). Participants were considered eligible for evaluation for oral treatment by BinaxNOW if they either: 1) initially tested positive, or 2) initially tested negative were expected to return within 5 days of symptom onset; and eligible by RT-PCR if their RT-PCR results returned within 5 days of symptom onset. Results are reported with 95% confidence intervals based on the bootstrapped samples.

### Ethics statement

The UCSF Committee on Human Research determined the study met criteria for public health surveillance. All participants provided electronic informed consent for dual testing; parental consent was obtained for participants under the age of 18.

## Results

We evaluated data from 25,309 visits between January 1 and June 26, 2022 with valid results from BinaxNOW. During the entire study period, 6,672 (26%) of visits had a positive Binax-NOW test (S1 Fig). During the January and June periods, 2,799 of these persons had valid concomitant RT-PCR results and BinaxNOW tests. Among these 2,799 persons, 50.5% were self-reported as female, 46.7% as male, and 2.8% as other (Table 1). In this population, 74.1% self-reported as Latinx or American Indian from Central or South America. There were 1,137/2,799 (41%) SARS-CoV-2 RT-PCR positive samples.

Of the 1,137 persons with positive RT-PCR, 708 (62%) reported at least one symptom associated with COVID-19. The most reported symptoms were cough (42.9%), sore throat (31.3%) and congestion (28.0%). As reported previously, loss of smell was rare (3.2%) [20]. Primary vaccine series only was reported by 31.1% overall and at least one booster by 40.8%. Among those RT-PCR positive, 39.1% had received the primary series only, 30.0% had one booster and 5.6% had 2 boosters; 24.2% reported prior infection; and 23.8% reported vaccination plus prior infection. Vaccine and prior infection status was by self-report.

We sequenced 720/6,672 (10.8%) of the specimens between January and June 30. Over 98% of the sequences were omicron, with BA.1 predominant in January, BA.2 in March, and BA.4 and BA.5 appearing in May. At the end of the study, BA.5 composed 33% of sequenced samples (Fig 1).

Among 1,137 RT-PCR positive samples, 69% were detected by BinaxNOW; 94% (669/711) of RT-PCR positive sample with Ct <30 were detected by BinaxNOW. BinaxNOW detected 66% (391/592) and 73% (397/545) of all RT-PCR positives in January and June, respectively (Fig 2). BinaxNOW detected over 90% of RT-PCR positive samples with Ct <30 across each sublineage (S2 Fig). When we examined BinaxNOW positivity stratified by time from symptom onset regardless of Ct, BinaxNOW positivity was 81% (158/196) in first 2 days compared to 78% (304/388) for days 3–10. However, for symptom onset days 1–2, BinaxNOW detection varied between January (72%; 74/103) and June (90%; 84/93) (p < 0.01; Fig 3). BinaxNOW detection of RT-PCR positive cases with Ct <30 was over 90% across strata of self-reported vaccine status or prior SARS-CoV-2 infection (S3 Fig).

In analyses of testing modality potential to identify persons eligible for treatment evaluation, BinaxNOW detected 81.7% (361/442, 95% CI: 78–85) of RT-PCR positive persons presenting within the 5-day window from symptom onset. In comparison, RT-PCR (24-hour turnaround) would have detected 84.1% of eligible persons within the 5-day window (372/442, 95% CI: 80–87) and RT-PCR (48-hour turnaround) would have detected 66.9% (296/442, 95% CI: 62–71) (Table 2). In simulations to estimate the percent of participants identified as eligible for oral treatment evaluation based on results of SARS-CoV-2 RT-PCR versus BinaxNOW, there was no difference by testing modality in January (Fig 4). In June, BinaxNOW would have identified 88.6% (95% CI 85–92%) while RT-PCR (average 24-hour turnaround) would have identified 73.8% (95% CI 69–79%), an absolute difference of 14.7% (95% CI: 9–21) favoring BinaxNOW. Simulations evaluating sensitivity to RT-PCR turnaround times and BinaxNOW repeat testing probability (S4 Fig) found that, in settings where RT-PCR results can be returned within hours, RT-PCR identifies a higher proportion of persons eligible for treatment evaluation. However, under a range of assumptions that reflect common settings in community sites, such as where our study was conducted, BinaxNOW identified as many or more persons than RT-PCR as eligible for treatment evaluation, due to the reduced turnaround time of rapid antigen tests.

## Discussion

We found that the BinaxNOW rapid antigen test detected 69% of RT-PCR positive SARS--CoV-2 anterior nasal swab specimens and 94% of specimens with Ct <30, among persons

**Table 1. Baseline demographics for participants, stratified by nasal swab RT-PCR result.**

| | | RT-PCR Positive | RT-PCR Negative | All |
|---|---|---|---|---|
| | | N = 1137 | N = 1662 | N = 2799 |
| **Sex** | **Female** | 518 (45.6%) | 895 (53.9%) | 1413 (50.5%) |
| | **Male** | 598 (52.6%) | 710 (42.7%) | 1308 (46.7%) |
| | **Other** | 21 (1.9%) | 57 (3.4%) | 78 (2.8%) |
| **Age (years)** | **< = 12** | 85 (7.5%) | 188 (11.3%) | 273 (9.8%) |
| | **13–17** | 82 (7.2%) | 96 (5.8%) | 178 (6.4%) |
| | **18–24** | 121 (10.6%) | 140 (8.4%) | 261 (9.3%) |
| | **25–34** | 225 (19.8%) | 290 (17.4%) | 515 (18.4%) |
| | **35–44** | 239 (21.0%) | 289 (17.4%) | 528 (18.9%) |
| | **45–54** | 198 (17.4%) | 252 (15.2%) | 450 (16.1%) |
| | **55–64** | 110 (9.7%) | 232 (14.0%) | 342 (12.2%) |
| | **65+** | 77 (6.8%) | 175 (10.5%) | 252 (9.0%) |
| **Race/Ethnicity** | **Hispanic/Latinx** | 806 (70.9%) | 1114 (67.0%) | 1920 (68.6%) |
| | **White/Caucasian** | 81 (7.1%) | 137 (8.2%) | 218 (7.8%) |
| | **American Indian from South or Central America** | 71 (6.2%) | 84 (5.1%) | 155 (5.5%) |
| | **Asian** | 56 (4.9%) | 95 (5.7%) | 151 (5.4%) |
| | **Black or African American** | 18 (1.6%) | 36 (2.2%) | 54 (1.9%) |
| | **American Indian or Alaska Native** | 0 (0.0%) | 7 (0.4%) | 7 (0.3%) |
| | **Pacific Islander or Native Hawaiian** | 2 (0.2%) | 5 (0.3%) | 7 (0.3%) |
| | **Other** | 103 (9.1%) | 184 (11.1%) | 287 (10.3%) |
| **Income** | **Less than 50k** | 516 (45.4%) | 705 (42.4%) | 1221 (43.6%) |
| | **50k – 100k** | 104 (9.1%) | 148 (8.9%) | 252 (9.0%) |
| | **More than 100k** | 48 (4.2%) | 64 (3.9%) | 112 (4.0%) |
| | **Refused** | 469 (41.2%) | 745 (44.8%) | 1214 (43.4%) |
| **Symptom Status** | **Symptomatic** | 708 (62.3%) | 486 (29.2%) | 1194 (42.7%) |
| | **Asymptomatic** | 426 (37.5%) | 1159 (69.7%) | 1585 (56.6%) |
| **Symptom (Among symptomatic patients)** | **Cough** | 488 (42.9%) | 239 (14.4%) | 727 (26.0%) |
| | **Sore Throat** | 356 (31.3%) | 190 (11.4%) | 546 (19.5%) |
| | **Congestion** | 318 (28.0%) | 187 (11.3%) | 505 (18.0%) |
| | **Headache** | 279 (24.5%) | 152 (9.1%) | 431 (15.4%) |
| | **Myalgia** | 228 (20.1%) | 121 (7.3%) | 349 (12.5%) |
| | **Fever** | 223 (19.6%) | 84 (5.1%) | 307 (11.0%) |
| | **Fatigue** | 180 (15.8%) | 102 (6.1%) | 282 (10.1%) |
| | **Shortness of Breath** | 74 (6.5%) | 38 (2.3%) | 112 (4.0%) |
| | **Nausea** | 44 (3.9%) | 27 (1.6%) | 71 (2.5%) |
| | **Diarrhea** | 34 (3.0%) | 27 (1.6%) | 61 (2.2%) |
| | **Loss of Smell** | 36 (3.2%) | 18 (1.1%) | 54 (1.9%) |
| | **Other** | 3 (0.3%) | 6 (0.4%) | 9 (0.3%) |
| **Vaccine Status (Self-Report)** | **Unvaccinated** | 101 (8.9%) | 137 (8.2%) | 238 (8.5%) |
| | **Started Primary Vaccine Series** | 102 (9.0%) | 139 (8.4%) | 241 (8.6%) |
| | **Completed Primary Vaccine Series** | 445 (39.1%) | 426 (25.6%) | 871 (31.1%) |
| | **Primary Vaccine Series with Booster** | 407 (35.8%) | 736 (44.3%) | 1143 (40.8%) |
| | **Refuse/Unknown** | 82 (7.2%) | 224 (13.4%) | 306 (10.9%) |
| **Prior Infection (Self-Report)** | **Yes** | 275 (24.2%) | 298 (17.9%) | 573 (20.5%) |
| | **No** | 726 (63.9%) | 1089 (65.5%) | 1815 (64.8%) |
| | **Refuse/Unknown** | 136 (11.9%) | 275 (16.5%) | 411 (14.7%) |

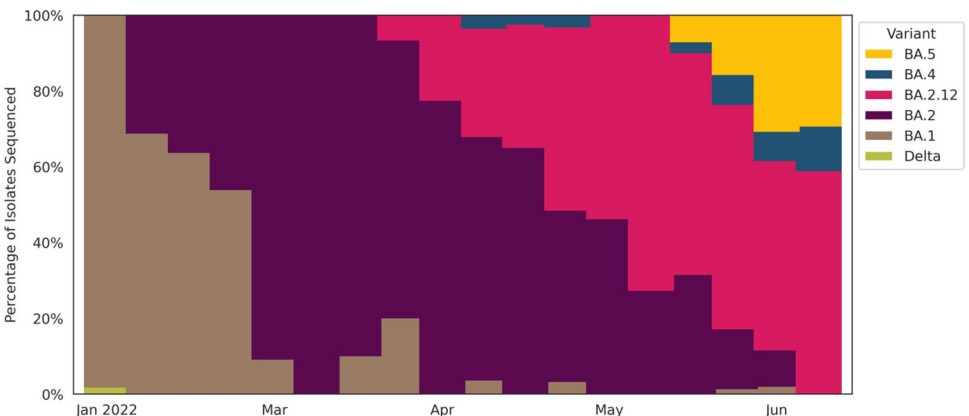

**Fig 1. SARS-CoV-2 Lineages of all RT-PCR or rapid antigen test (BinaxNOW) positive participants sequenced between January 1 and June 26, 2022.** Each vertical bar corresponds with one week. The height of each color segment within the bar corresponds with the percent of participants in that week whose sequences were identified as that color's lineage. Abbreviations: Ct, cycle threshold; RT-PCR, reverse transcriptase polymerase chain reaction.

presenting in the setting of high rates of ongoing community transmission and evolving omicron variants. BinaxNOW—even with lower sensitivity than RT-PCR—detected infection among symptomatic persons within the window of five days from symptom onset for treatment eligibility in as many or more cases as RT-PCR assuming a 24- or 48-hour turnaround for PCR results.

These data support use of BinaxNOW at community sites as a compliment to RT-PCR testing entry points for identification of individuals in need of treatment. Many health systems now permit results from purchased home COVID rapid antigen tests as an entry point for treatment; some health systems provide these tests free for their members. For persons uninsured or unable to navigate the health systems, access to rapid antigen tests can be difficult and prohibitively expensive. Although government distribution programs of free rapid antigen tests can be helpful, these programs often provide insufficient tests for very large immigrant households that are common in the community we serve. Free community testing sites such as Unidos en Salud using proven rapid antigen testing such as BinaxNOW can mitigate potential disparities to treatment access fueled by economic and health system considerations.

Among the many challenges of the public health response to COVID-19 is the need to ensure that tools and guidelines are suitable for rapidly shifting virus and host immunity [21]. To date, SARS-CoV-2 antigen assays, despite lower sensitivity than RT-PCR, have proven an important resource for rapid detection of persons with high levels of virus [1]. However, there is heterogeneity in assays, and there have been mixed reports on changes of rapid antigen performance for detection of omicron variants from in vitro and clinical studies [21–27]. There was even formal recommendations against use of certain antigen tests for omicron detection by the US FDA in 2022 [28]. Together, these reports emphasize the important role of in vitro and field testing for detection of SARS-CoV-2.

Our data can inform the public health utility of only a single assay—BinaxNOW—as assessed in in a community-based setting during recent omicron surges. Although there are many genetic mutations in omicron variants, we showed consistently high detection (>90%) for persons with high levels of virus across omicron sublineages BA.1, BA.2, BA.2.12, and BA.5, using BinaxNOW in RT-PCR positive samples. The detection levels reported here are similar to those we reported in this community setting when prior SARS-CoV-2 variants were dominating, providing reassurance for use of these tests in the current era [10, 12, 29, 30].

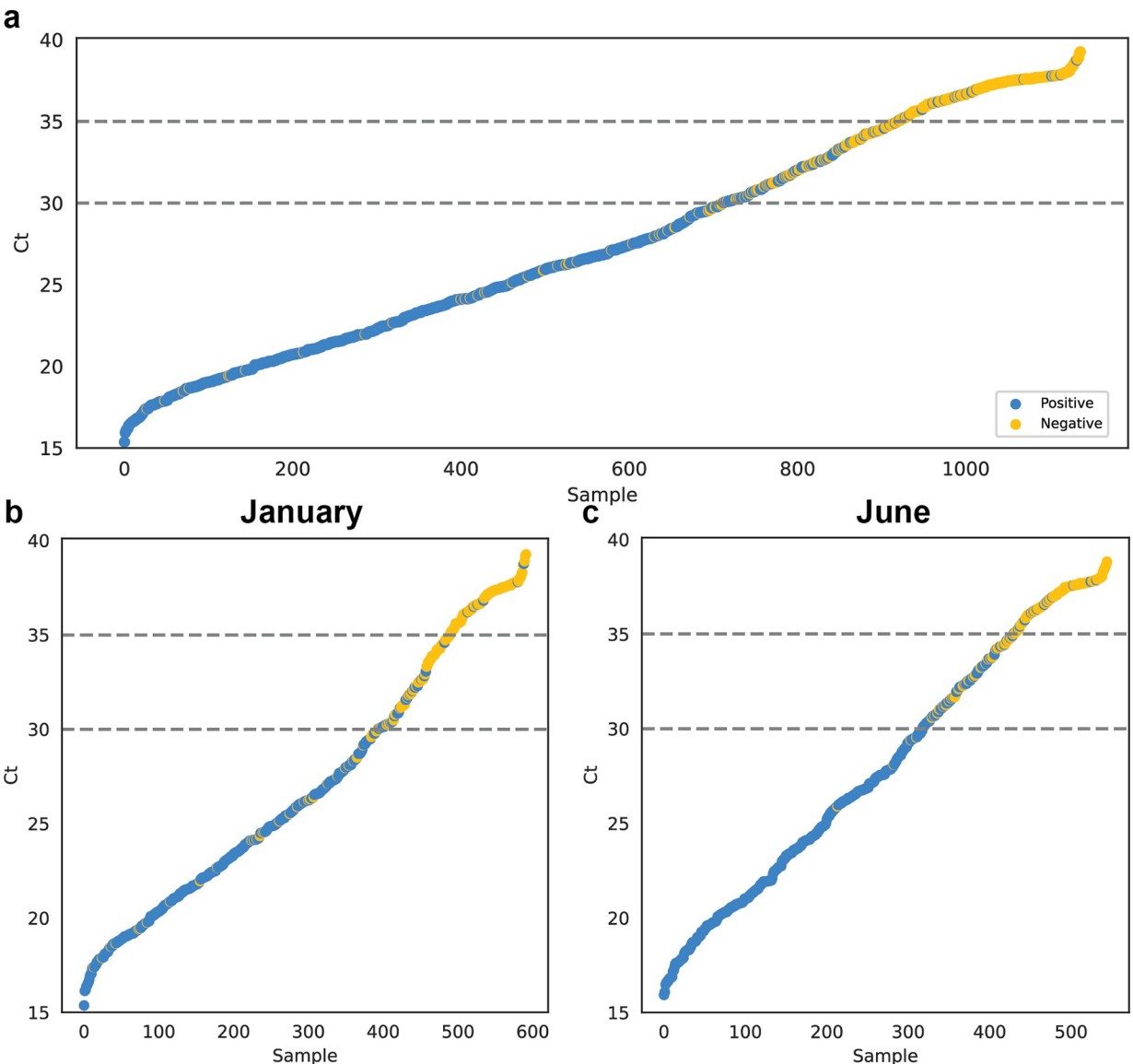

**Fig 2.** SARS-CoV-2 RT-PCR Ct values and BinaxNOW rapid antigen test results of all RT-PCR positive participants tested between January 1 and June 26, 2022 (panel a) and stratified according to January or June test date (panels b and c). Average viral Ct values of all individuals with positive RT-PCR test results (N = 1321) are plotted in ascending order of Ct value. Each point represents one individual. Blue circles are individuals whose samples were positive on both rapid antigen test (BinaxNOW) and on RT-PCR test. Orange circles represent individuals who were RT-PCR positive but rapid antigen test negative. Abbreviations: Ct, cycle threshold; RT-PCR, reverse transcriptase polymerase chain reaction.

Omicron includes mutations in the nucleocapsid protein (NP), the target of BinaxNOW, but this genetic variability did not appear to affect field performance of BinaxNOW.

One of the interesting observations in our data was the lower proportion of RT-PCR samples that were BinaxNOW positive within the first 2 days of symptom onset in January compared to June. This appears to be driven by a decrease in viral load at time of testing among persons with recent symptom onset. It has been pointed out previously that rapid antigen assays detect fewer cases than RT-PCR during the upswing of the virus (when viral levels are not high enough for rapid antigen detection), and that this is more likely to be seen among testers at the beginning of a surge such as occurred in January [8]. However, this seems unlikely

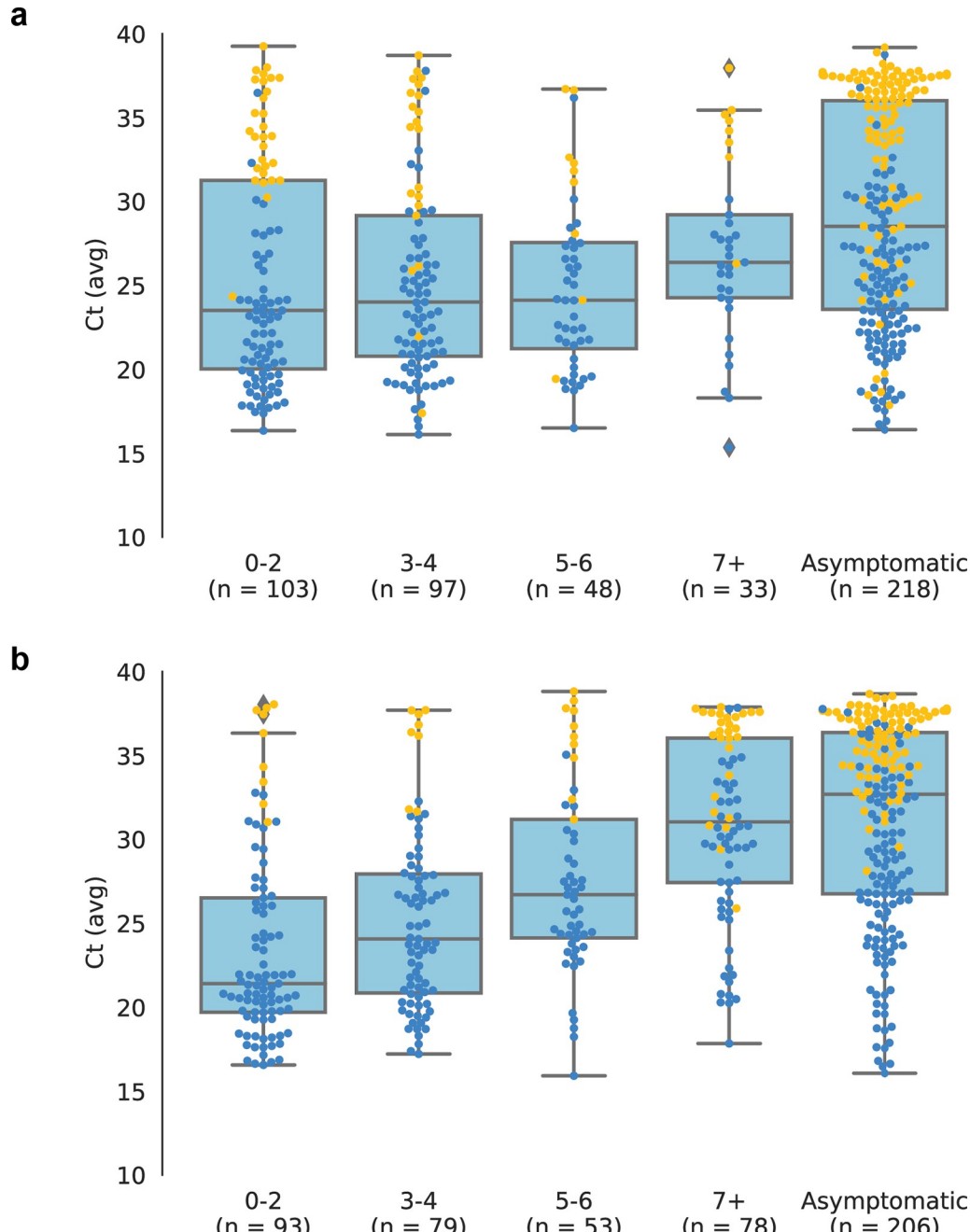

**Fig 3.** SARS-CoV-2 RT-PCR Ct values and BinaxNOW rapid antigen test results of participants tested between January 1 and June 26, 2022, stratified by days since symptom onset for the January (a) and June (b) periods. Box plot shows first quartile, median, and third quartile in the shaded region; diamonds indicate outliers beyond 1.5 times the interquartile range. Blue circles overlaid are individuals whose samples were positive on both rapid antigen test (BinaxNOW) and on RT-PCR test. Orange circles represent individuals who were RT-PCR positive but rapid antigen test negative. Abbreviations: Ct, cycle threshold; RT-PCR, reverse transcriptase polymerase chain reaction.

to alone account for the apparent change in the relationship between viral load and symptom onset we observed. It is possible that the viral load at which symptom onset occurs may have shifted lower due to increasing immunity concurrent with the emergence of Omicron. Intriguingly, we observed the opposite temporal pattern here, with an evolution towards higher viral

**Table 2. Results of analysis comparing BinaxNOW to RT-PCR test under multiple assumptions.**

| Testing Method | Assumption | Positive within 5 days (%) | 95% CI | Mean Time to Positive (days) |
|---|---|---|---|---|
| BinaxNOW | Return for repeat testing | 90.3% | 87–93% | 3.1 |
| | No return for repeat testing | 81.7% | 78–85% | 2.9 |
| PCR | 24hr turnaround | 84.2% | 80–87% | 3.5 |
| | 48hr turnaround | 67.0% | 62–71% | 4.1 |
| | 72hr turnaround | 44.3% | 40–49% | 4.7 |

loads immediately following symptom onset between the omicron surge in January and six months later.

There are many possible reasons for these observations. Testing behavior may have changed such that persons reported symptoms and sought testing earlier in January than June. Reinfection, previously an unusual occurrence, was being documented around the world, including among vaccinated persons who may have had lower index of suspicion of COVID-19 [31, 32]. These individuals may have perceived symptoms differently and delayed testing in June compared to January. The variants present in June may have elicited a pace of inflammatory responses and clinical symptoms in susceptible hosts different than in January. In an elegant

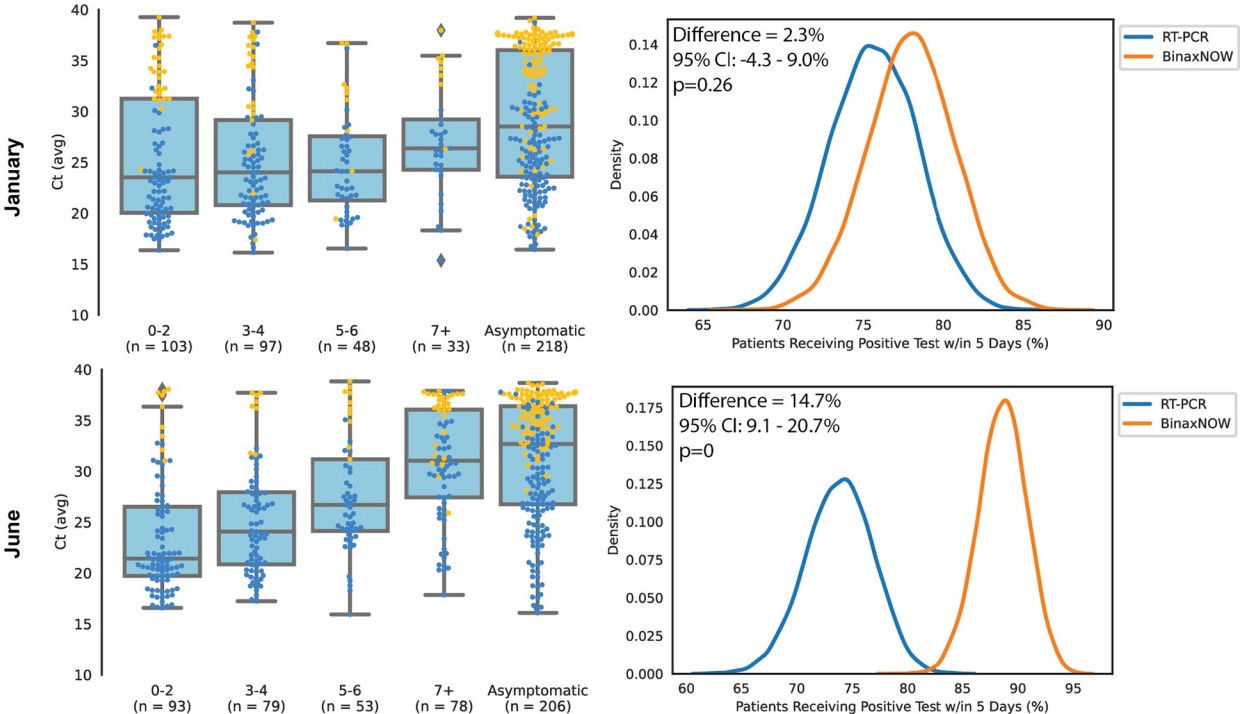

**Fig 4. Results of Monte Carlo simulations to estimate the expected percent of participants that would be eligible for oral treatment evaluation based on results of SARS-CoV-2 RT-PCR or rapid antigen test (BinaxNOW) testing under different scenarios.** Each row corresponds to a different scenario; the top row uses conditions from January 2022 and the bottom row uses conditions from June 2022. The left column shows the distribution of SARS-CoV-2 RT-PCR Ct by days since symptom onset used as input into the simulations; orange dots correspond to participants who were RT-PCR positive but rapid antigen test negative and blue dots correspond to participants who were both RT-PCR and rapid antigen test positive. The right column show the resulting distributions of percent of participants who would be eligible for oral treatment evaluation based on RT-PCR (blue) versus rapid antigen test (orange) testing. These simulations assume 15% of participants with false negative rapid antigen tests will return and test positive three days after their initial test, and that RT-PCR test results will return after an average of 1 day. The mean difference, 95% confidence interval, and p-value are overlaid on the respective graphs, all calculated using the Monte Carlo simulations. Abbreviations: Ct, cycle threshold; RT-PCR, reverse transcriptase polymerase chain reaction.

modeling exercise using a large data set from England, during the initial days from symptoms onset, variant status had the largest influence on variations in Ct threshold [33]. Finally, it is probable that a combination of factors contributed.

One pillar of the COVID-19 response is oral antimicrobial treatments for individuals at high risk for progression to serious disease [9]. These treatments must be started within 5 days of symptom onset. The fastest entry into identifying persons with suspected COVID-19 for treatment would be a highly sensitive RT-PCR or nucleic acid -based assay with turnaround within hours. However, this is not accessible in many settings and requires a health professional to connect with a patient to convey results and evaluate for treatment. We were interested in evaluating whether, in a community field setting with existing infrastructure to connect with patients, BinaxNOW could be a public health tool for treatment evaluation even considering its lower sensitivity than RT-PCR. Our data and simulations, across varying patient presentation and BinaxNOW detection profiles, and plausible RT-PCR turnaround times, show that BinaxNOW identified as many or more persons within their treatment eligibility window than RT-PCR. Our simulations are not intended to imply that BinaxNOW is a better test than RT-PCR for SARS-CoV-2 detection, but rather to show, using real world data in a community setting, that under a variety of scenarios it is a viable entry point into treatment evaluation.

Our community testing and vaccination site extends the reach of traditional health systems in a public health response to COVID-19 by offering walk-up services that address some barriers that persons who are uninsured, monolingual, or managing multiple jobs may experience. Although we have shown that BinaxNOW is a reasonable entry point for treatment evaluation in this setting, its impact will require immediate treatment access. Oral treatments for COVID were initially limited, but now that access has increased, we have started a rapid screening and dispensing program for nirmatrelvir/ritonavir for eligible symptomatic individuals testing positive using BinaxNOW. If this approach works, it provides another point-of entry for a community that continues to be disproportionately affected by COVID.

We recognize the limitations of these surveillance data. First, results from our study may not apply to all rapid antigen tests, and home testing is subject to variability in user performance. Second, the comparative performance of these two testing modalities for treatment evaluation will depend on viral burden and time since symptom onset at time of testing, which may vary across populations, geographical regions, and over time due to factors such as changes in testing behavior, immune landscape, and epidemic context. This underscores the importance of both in vitro and field studies for evolving conditions in various settings. Further, we can not distinguish between biological and behavioral explanations for viral load differences at time of testing based on these serial cross-sectional data of persons presenting for testing. Third, our measures of BinaxNOW positivity in persons with SARS-CoV-2 on RT-PCR are confined to disease detection by anterior nasal swabs. Studies have shown obtaining repeated specimens from multiple sites, including pharynx, can increase disease detection [27]. However, this is not feasible at high volume community sites.

In conclusion, BinaxNOW detected persons with high levels of SARS-CoV-2 infection in multiple omicron sublineages within a time frame enabling treatment evaluation. Community sites, such as Unidos en Salud, play an important role in the public health response by providing free rapid testing for low-income populations that can reduce disparities in treatment access.

## Supporting information

**S1 Fig. SARS CoV-2 BinaxNOW rapid antigen test statistics between January 1 and June 26, 2022 at walk-up community testing site Unidos en Salud in the Mission District in San**

**Francisco, California.** The blue line indicates the test positivity over time, corresponding with the left axis. The yellow line indicates the number of tests performed weekly, corresponding with the right axis.
(TIF)

**S2 Fig. BinaxNOW positivity among SARS-CoV-2 RT-PCR positive participants with Ct < 30, with confirmed lineage data.** Each bar represents the BinaxNOW positivity rate among the corresponding lineage and related sublineages, with 95% confidence intervals. The grey dashed line corresponds with 100% of SARS-CoV-2 RT-PCR positive participants with Ct < 30 also testing positive on BinaxNOW.
(TIF)

**S3 Fig.** SARS-CoV-2 RT-PCR Ct values and BinaxNOW rapid antigen test results (panel a), and BinaxNOW rapid antigen test detection (panel b) for participants tested between January 1 and June 26, 2022, stratified by months since prior self-reported vaccine or infection status. Panel A shows box plot shows first quartile, median, and third quartile in the shaded region; diamonds indicate outliers beyond 1.5 times the interquartile range. Blue circles overlaid are individuals whose samples were positive on both rapid antigen test (BinaxNOW) and on RT-PCR test. Orange circles represent individuals who were RT-PCR positive but rapid antigen test negative. Panel B shows bar plots with percent of RT-PCR positive participants with SARS-CoV-2 RT-PCR Ct less than 30 who were also BinaxNOW positive. Error bars indicate 95% confidence interval for each estimate. The grey dashed line corresponds with 100% of SARS-CoV-2 RT-PCR positive participants with Ct < 30 also testing positive on BinaxNOW. Abbreviations: Ct, cycle threshold; RT-PCR, reverse transcriptase polymerase chain reaction.
(TIF)

**S4 Fig. Results of analysis evaluating Monte Carlo simulation sensitivity to underlying assumptions.** Rows correspond to variations of the expected time to receive RT-PCR results in days, as modeled by an exponentially distributed random variable; the rows correspond with one hour, half day, full day, day and a half, or two days. The columns correspond with the expected percent of patients returning and testing positive after a false negative, as modeled by a binomially distributed random variable; the columns correspond with 0%, 15%, 25%, 50%, 75%, and 100%. Each cell corresponds with the expected difference between the percent of participants being eligible for oral treatment based on rapid antigen test minus those eligible based on RT-PCR; a positive number indicates rapid antigen test identifying more participants, and a negative number indicated RT-PCR identifying more. Abbreviations: Ct, cycle threshold; RT-PCR, reverse transcriptase polymerase chain reaction.
(TIF)

## Acknowledgments

The authors thank the community for their extraordinary support and ongoing partnership.

**Disclaimer:** The findings and conclusions in this article are those of the authors and do not necessarily represent the views opinions of the California Department of Public Health or the California Health and Human Services Agency.

## Author Contributions

**Conceptualization:** John Schrom, Carina Marquez, Chung-Yu Wang, Genay Pilarowski, Douglas Black, Maria G. Contreras Oseguera, Aleks Zavaleta, Jacqueline Martinez, Carol Glaser, Kathy Jacobson, Maya Petersen, Diane V. Havlir.

**Data curation:** John Schrom, Carina Marquez, Chung-Yu Wang, Aditi Saxena, Anthea M. Mitchell, Salu Ribeiro, Genay Pilarowski, Susana Rojas, Maria G. Contreras Oseguera, Edgar Castellanos Diaz, Susy Rojas, Jacqueline Martinez, Maya Petersen, Joseph DeRisi.

**Formal analysis:** John Schrom, Carina Marquez, Chung-Yu Wang, Aditi Saxena, Anthea M. Mitchell, Genay Pilarowski, Douglas Black, Jacqueline Martinez, Gabriel Chamie, Maya Petersen, Joseph DeRisi, Diane V. Havlir.

**Funding acquisition:** Diane Jones, Kathy Jacobson.

**Investigation:** John Schrom, Carina Marquez, Chung-Yu Wang, Anthea M. Mitchell, Susana Rojas, Douglas Black, Maria G. Contreras Oseguera, Edgar Castellanos Diaz, Diane Jones, Valerie Tulier-Laiwa, Aleks Zavaleta, Jacqueline Martinez, Gabriel Chamie, Kathy Jacobson, Joseph DeRisi, Diane V. Havlir.

**Methodology:** John Schrom, Carina Marquez, Chung-Yu Wang, Anthea M. Mitchell, Genay Pilarowski, Susana Rojas, Douglas Black, Edgar Castellanos Diaz, Susy Rojas, Diane Jones, Jacqueline Martinez, Gabriel Chamie, Kathy Jacobson, Maya Petersen, Joseph DeRisi, Diane V. Havlir.

**Project administration:** Salu Ribeiro, Susana Rojas, Douglas Black, Maria G. Contreras Oseguera, Edgar Castellanos Diaz, Joselin Payan, Susy Rojas, Diane Jones, Carol Glaser, Diane V. Havlir.

**Resources:** Salu Ribeiro, Robert Nakamura, Susana Rojas, Douglas Black, Joselin Payan, Susy Rojas, Diane Jones, Aleks Zavaleta, Carol Glaser, Kathy Jacobson, Joseph DeRisi.

**Software:** Jacqueline Martinez.

**Supervision:** Genay Pilarowski, Susana Rojas, Joselin Payan, Susy Rojas, Diane Jones, Valerie Tulier-Laiwa, Joseph DeRisi, Diane V. Havlir.

**Validation:** John Schrom, Jacqueline Martinez, Maya Petersen.

**Visualization:** John Schrom, Jacqueline Martinez.

**Writing – original draft:** John Schrom, Carina Marquez, Maya Petersen, Diane V. Havlir.

**Writing – review & editing:** John Schrom, Carina Marquez, Gabriel Chamie, Maya Petersen, Joseph DeRisi, Diane V. Havlir.

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
