## [Decision Letter · Decision Letter 0]

19 Dec 2022

PONE-D-22-30851Field assessment of BinaxNOW antigen tests as COVID-19 treatment entry point at a community testing site in San Francisco during evolving omicron surgesPLOS ONE

Dear Dr. Havlir,

Thank you for submitting your manuscript to PLOS ONE. After careful consideration, we feel that it has merit but does not fully meet PLOS ONE’s publication criteria as it currently stands. Therefore, we invite you to submit a revised version of the manuscript that addresses the points raised during the review process. I could obtain only one reviewer's comments.  

We look forward to receiving your revised manuscript.

Kind regards,

Etsuro Ito

Academic Editor

PLOS ONE

Journal Requirements:

2. Please ensure that you have discussed whether all data were fully anonymized before you accessed them.

Reviewers' comments:

Reviewer's Responses to Questions

**Comments to the Author**

1. Is the manuscript technically sound, and do the data support the conclusions?

Reviewer #1: Yes

2. Has the statistical analysis been performed appropriately and rigorously? 

Reviewer #1: Yes

3. Have the authors made all data underlying the findings in their manuscript fully available?

Reviewer #1: Yes

4. Is the manuscript presented in an intelligible fashion and written in standard English?

Reviewer #1: Yes

5. Review Comments to the Author

Reviewer #1: In this manuscript, Schrom et al performed simultaneous BinaxNOW and PCR testing on individuals presenting for testing at a community site in San Francisco. An impressive number of tests were performed- 25,309, with 2799 simultaneously being tested by PCR; sequencing demonstrated that the strains tested represented 5 omicron subvariants in circulation this year. To my reading, the manuscript meets the criteria for publication at PLoS One; changing framing somewhat could improve the readability and potentially the impact of the paper.

The primary knowledge gap that the manuscript is seeking to address is not clear. In some places, it sounds like the manuscript is looking to ask whether RATs (specifically BinaxNOW) can be useful as initial tests for treatment initiation from a timing standpoint. In other places it sounds like the manuscript is looking to ask whether BinaxNOW can readily detect the omicron variants that have been in circulation this year. Clinical practice in COVID-19 has changed rapidly, and in many (if not most) places, home RATs are the test that most commonly leads to treatment initiation. The framing of the manuscript has not been changed to accommodate this reality, leaving the reader to wonder why the work was performed. Some reframing to acknowledge this practical reality and contextualize the results within it would substantially improve the manucript.

6. PLOS authors have the option to publish the peer review history of their article (what does this mean?). If published, this will include your full peer review and any attached files.

Reviewer #1: No

---

## [Author Response · Author response to Decision Letter 0]

11 Jan 2023

We appreciate the reviewer’s comment and have updated the framing of the manuscript to address contemporary issues in the field. We agree with the reviewer that availability of home tests has changed the landscape of entry into treatment for many persons. However, low-income community members still have limited access to over-the-counter tests because of their high cost; in addition, many high-risk individuals lack agency to obtain free tests, and the allotted number can be insufficient for large households. Finally, rigorous data using specific rapid tests for detection of the evolving variants are lacking; in the absence of these data, anecdotal experiences in lay press and social media lead to misinformation and confusion for the public. In the revised submission, we more clearly state our knowledge gaps and implications of our findings.

---

## [Editor Report · Decision Letter 1]

13 Mar 2023

Field assessment of BinaxNOW antigen tests as COVID-19 treatment entry point at a community testing site in San Francisco during evolving omicron surges

PONE-D-22-30851R1

Dear Dr. Havlir,

We’re pleased to inform you that your manuscript has been judged scientifically suitable for publication and will be formally accepted for publication once it meets all outstanding technical requirements.

Kind regards,

Etsuro Ito

Academic Editor

PLOS ONE

---

## [Editor Report · Acceptance letter]

16 Mar 2023

PONE-D-22-30851R1 

Field assessment of BinaxNOW antigen tests as COVID-19 treatment entry point at a community testing site in San Francisco during evolving omicron surges 

Dear Dr. Havlir:

I'm pleased to inform you that your manuscript has been deemed suitable for publication in PLOS ONE. Congratulations! Your manuscript is now with our production department. 

Kind regards, 

on behalf of

Prof. Etsuro Ito 

Academic Editor

PLOS ONE